# Dietary Supplementation of Chestnut Tannins in Prepartum Dairy Cows Improves Antioxidant Defense Mechanisms Interacting with Thyroid Status

**DOI:** 10.3390/metabo13030334

**Published:** 2023-02-24

**Authors:** Radiša Prodanović, Sreten Nedić, Ivan Vujanac, Jovan Bojkovski, Svetlana Nedić, Ljubomir Jovanović, Danijela Kirovski, Sunčica Borozan

**Affiliations:** 1Department of Ruminants and Swine Diseases, Faculty of Veterinary Medicine, University of Belgrade, 11000 Belgrade, Serbia; 2Department of Reproduction, Fertility and Artificial Insemination, Faculty of Veterinary Medicine, University of Belgrade, 11000 Belgrade, Serbia; 3Department of Physiology and Biochemistry, Faculty of Veterinary Medicine, University of Belgrade, 11000 Belgrade, Serbia; 4Department of General Education, Faculty of Veterinary Medicine, University of Belgrade, 11000 Belgrade, Serbia

**Keywords:** antioxidant status, chestnut tannin, dairy cows, thyroid hormones

## Abstract

Cows in the peripartal period undergo changes in thyroid hormones and are susceptible to lipomobilization and/or oxidative stress. The addition of chestnut tannins as polyphenolic compounds in the diet may improve feed efficiency and prevent oxidative stress-related health disorders in transition cows. However, the relationship between chestnut tannin supplementation and thyroid function, which plays an important role in metabolic regulation, has not been investigated in dairy cows. This study was conducted to investigate the effects of chestnut tannin supplementation during the close-up period on thyroid status and to evaluate the interaction between thyroid hormones and oxidative stress biomarkers in prepartum dairy cows. Forty multiparous Holstein cows were fed either a diet containing chestnut tannins (CNTs, *n* = 20, 1.96 g chestnut tannins/kg feed, dry matter) or a non-supplemented diet (CON, *n* = 20) during the last 25 ± 2 days of gestation. Blood samples were collected on the first day of study (before chestnut tannin supplementation) and d 5 before parturition to measure hormonal and oxidative stress indices. Serum concentrations of T3 (*p* = 0.04) and T4 (*p* = 0.05) were higher in CNT cows than in the CON group on day 5 before parturition. Thyroid status of CNT cows was associated with higher serum total antioxidant capacity (T-AOC, *p* < 0.01), activities of superoxide dismutase (SOD, *p* = 0.03) and glutathione peroxidase (GPx, *p* = 0.01), and reduced glutathione concentration (GSH, *p* = 0.05). Serum thiobarbituric acid reactive substances (TBARS) were lower (*p* = 0.04) which was associated with lower aspartate aminotransferase (AST, *p* = 0.02), and lactate dehydrogenase (LDH, *p* = 0.01) activities in the CNT than in the CON group. Estradiol and progesterone did not differ between CNT and CON cows. Chestnut tannin supplementation improves antioxidant protection, prevents oxidation-reduction processes, reduces the degree of liver cell membrane damage, and protects thyroid tissue from damage, allowing higher T3 and T4 synthesis. Considering the importance of the thyroid hormone status before parturition, mechanisms of thyroid hormone regulation in CNT-supplemented dairy cows require more detailed investigations.

## 1. Introduction

Oxidative stress is caused by the imbalance between the production of reactive oxygen species (ROS) and the neutralization capacity of antioxidant mechanisms [1]. Nutrient metabolism and inflammation in the transition period in dairy cows are associated with oxidative stress, which underlies the pathogenesis of various diseases during this period [1,2,3]. The transition period, as a period of high metabolic demands on the dairy cow, increases the production of ROS, which increases the need for antioxidants [4,5]. The latter may be endogenously produced by the organism and/or derived from the diet [6]. Under the influence of ROS, impaired hepatocyte cell membrane integrity may lead to increased extracellular aspartate aminotransferase (AST) and lactate dehydrogenase (LDH) activities. Thus, the blood plasma activities of AST and LDH can be used as markers of ROS-induced liver cell injury in cattle [3]. Despite advances in nutritional management during the transition period, oxidative stress remains a problem for immune system impairment and health in transition cows.

The ability of tannins to improve feed efficiency and antioxidant status in dairy cows is well known, and they have often been supplemented in the diet of dairy cows [7,8,9]. Tannins have also been suggested as effective alternatives to antibiotics to suppress rumen methanogenesis through their antimicrobial activity [10,11]. Extracts of both condensed and hydrolysable tannins can be incorporated into dairy cows’ diet, whereas the quality and quantity of their effects may differ depending on the tannin source [7,8,12]. The use of condensed tannins in the ruminant diet due to its protein precipitation and anti-microbial and anti-parasitic activities has been widely researched [12,13,14]. However, compared to condensed tannins, hydrolysable tannins have a greater biological activity especially in regards to their antioxidant properties [8,15]. Hydrolysable tannins from sweet chestnuts are effective at scavenging ROS and at protecting liver and kidney tissues against oxidants as well as liposomes from lipid peroxidation [7,8]. Previous studies in transition cows have shown that total antioxidant capacity (T-AOC), superoxide dismutase (SOD), glutathione peroxidase (GPx), and paraoxonase 1 (PON 1) activity were increased in response to chestnut tannin supplementation, and lipid peroxidation was decreased, resulting in lower oxidative stress [7,9,16].

It is often suggested that oxidative stress is involved in the impairment of thyroid function, as evidenced by a decrease in thyroid hormones, which in turn may increase oxidative stress [17]. The relationship between thyroid hormones and oxidative stress biomarkers has been studied in humans and laboratory rodents, and a reciprocal relationship between thyroid status and oxidative stress has been found [18,19]. Thyroid hormones have been shown to regulate the antioxidant defence system, at least in part, by stimulating the transcription factor peroxisome proliferator-activated receptor gamma-coactivator 1-alpha and the synthesis of antioxidant enzymes [6,20], which is crucial for the adaptation of animals to environmental and nutritional stress and may have a protective effect against oxidants in liver and kidney tissues [21]. Previous studies have shown that the shift in substrate utilization mediated by a decline in T3 and T4 in heat-stressed cows is associated with decreased antioxidant defence [22], suggesting a functional link between thyroid hormones and the antioxidant system. Thyroid status depends on sex hormone imbalance [23], which may also affect thyroid hormone production and function during lactation in dairy cows. Since both estrogen and progesterone are involved in regulating the amount of protein that carries thyroid hormones in the blood, such effects could have an important impact on the level of free thyroid hormones.

The adaptation of thyroid hormone metabolism is important during the transition from late gestation to early lactation because it regulates important metabolic and oxidative processes, and it has been shown that hypothyroidism in the close-up period in dairy cows may be an early indicator of metabolic disorders after parturition [24]. A better understanding of the nutritional factors affecting thyroid function in the close-up period can help to improve thyroid activity and prevent oxidative stress-related disorders [25,26].

Therefore, we hypothesized that chestnut tannin supplementation in the close-up diet can affect thyroid function through interaction with antioxidant defence mechanisms. The aim of this study is to investigate the effects of dietary chestnut tannin supplementation in the close-up diet on thyroid status (assessed with blood T3 and T4 concentrations) and antioxidant defence mechanisms (assessed with total antioxidant capacity, superoxide dismutase and glutathione peroxidase activities, and blood levels of thiobarbituric acid reactive substances, and reduced glutathione). In addition, blood estrogen and progesterone concentrations and LDH and AST activities were evaluated.

## 2. Materials and Methods

This study was approved by the Ethics Committee of the Faculty of Veterinary Medicine, University of Belgrade, under approval number 05/2015, in accordance with the National Regulation on Animal Welfare. In brief, forty late pregnant Holstein cows were ranked by parity and body condition score in descending order and alternately divided into two groups: the control group (CON, *n* = 20), which was not supplemented, and the group supplemented with chestnut tannins (CNTs, *n* = 20). The CNT cows received 20 g/d (1.96 g chestnut tannins/kg of diet, dry matter basis) of a commercially available product containing chestnut tannins (Tanimil SCC, Tanin Sevnica, Slovenia) during the last 25 ± 2 d of pregnancy. Tanimil SCC contains 48% hydrolysable and 2.1% condensed tannins originating from sweet chestnut extract. No additional tannin source was used other than sweet chestnut. Ten grammes of the product were mixed twice daily with 50 g of concentrate for the total mixed ration (TMR) and administered to each CNT cow immediately before morning and evening TMR feeding. Diets formulated to meet or exceed NRC (2001) requirements were fed in two equal portions at 6:30 am and 5:30 pm. Ingredients and chemical composition of the far-off and close-up diets are shown in Table 1. Cows were clinically examined for general condition, and only healthy cows without a history of metabolic disorders in the previous lactation were selected for the study. The clinical visit was carried out on the day before the first blood sampling (25 days before expected parturition), and cows were clinically examined on each day of study by general condition as well as by signs of damage to the gut, kidney, and liver. The animals did not exhibit any clinical health problems or signs of tannin toxicity during the trial period. Feed intake was monitored each day due to possible antinutritional effects of tannins and no significant difference in average DMI (mean ± SEM) was observed between the CON (9.9 ± 0.06 kg of DM) and CNT groups (9.8 ± 0.05 kg of DM).

### 2.1. Blood Samples and Analyses

Blood samples were collected from each cow on the first day of chestnut tannin supplementation (meaning on day 25 before expected calving) and then every day until calving, and blood obtained on day 5 before calving was used for the analysis. Blood was collected before the morning feeding by puncturing the jugular vein into Vacutainer blood collection tubes (Becton Dickinson, Plymouth, UK) with a clot activator for serum separation. The tubes were immediately placed in an ice box and transferred to the laboratory within one hour.

### 2.2. Metabolite and Hormone Analyses

Samples for the determination of aspartate aminotranspherase (AST), lactate dehydrogenase (LDH), triiodothyronine (T3), thyroxine (T4), estradiol (E2), and progesterone were collected in gel-coated blood tubes (Becton Dickinson, Plymouth, UK), centrifuged at 1800× *g* for 10 min, and aliquoted into 2 mL microfuge tubes. Aliquots of serum were stored at −20 °C until later analysis. Biochemical metabolites were analysed by the Department of Ruminants and Swine Diseases (Belgrade, Serbia) using the appropriate kits: AST (IFCC method) and LDH (pyruvate method) from BioSystems S.A. (Barcelona, Spain). Analyses were performed automatically using spectrophotometry (A15; Bio-Systems S.A., Barcelona, Spain). Serum levels of T3 and T4 were measured with a radioimmunoassay kit (INEP-Zemun, Serbia) validated for use with bovine serum. The mean intraassay coefficients of variation (CV) for duplicate samples were 3.7% and 4.2% for T3 and T4, respectively. Inter-assay CVs were less than 10%. Serum E2 and progesterone concentrations were determined photometrically with an automated immunoassay analyser (AIA 360, Tosoh, Japan) using a commercially available competitive enzyme immunoassay kit from the same manufacturer.

### 2.3. Estimation of Total Antioxidant Capacity in Blood Serum

Total antioxidant capacity (T-AOC) was evaluated using the ferric reducing antioxidant power method (FRAP). The reducing power was determined according to the method of Oyaizu [27]. Briefly, 40 µL of the sample was mixed with 200 mmol/L sodium phosphate buffer (pH 6.6) and 10 mg/mL potassium ferricyanide, and the mixture was incubated at 500 °C for 20 min. Then, 2.5 mL of 100 mg/mL trichloroacetic acid was added, and then the mixture was centrifuged at 2000× *g* for 10 min. The upper layer was mixed with 0.5 mL of deionized water and 1 mL of 1 mg/mL ferric chloride, and absorbance was measured at 700 nm against a blank. A higher absorbance indicated a higher reducing power. Different concentrations of butylhydroxytoluene were used as standards for calibration, and the results were expressed in micrograms of butylhydroxytoluene (BHT Eq) per mL of sample.

### 2.4. Estimation of Superoxide Dismutase (SOD) Activity in Blood Serum

The activity of superoxide dismutase (SOD) in serum was determined using the method based on the inhibition of autoxidation of epinephrine. The change in extinction was measured in time (6 min) at a wavelength of 480 nm [28]. The activity of SOD was expressed as U/mL of serum.

### 2.5. Estimation of Glutathione Peroxidase (GPx) Activity in Blood Serum

GPx activity was determined using the oxidation of GSH with GPx in a coupled test system with glutathione reductase. The decrease in absorbance at 340 nm as a result of NADPH + H+ used is continuously registered spectrophotometrically [29]. GPx activity was expressed as U/mL of serum.

### 2.6. Determination of Lipid Peroxidation (TBARS)

Lipid peroxidation was determined in serum by measuring thiobarbituric acid reactive substances (TBARS) according to the methods developed by Gutteridge [30] and Traverso et al. [31]. The assay evaluated the formation of a coloured adduct after the stoichiometric reaction between thiobarbituric acid (TBA) and various lipid-derived aldehydes, including malondialdehyde (MDA). The TBARS content released in the samples was measured at 535 nm. Results were expressed in nmol/mL of serum.

### 2.7. Determination of Reduced Glutathione (GSH)

The concentration of reduced GSH was determined spectrophotometrically at 412 nm using 5,5′-dithiobis-2-nitrobenzoic acid (DTNB) (Ellman’s reagent). The concentration of GSH was presented as µmol/mL of serum [32].

### 2.8. Statistical Analyses

Data were analysed using the Statistica v.8 commercial software (Stat Soft, Inc., Tulsa, OK, USA) and presented as mean ± SE (standard error) for all examined indices. The normality of the residuals was tested using the Shapiro–Wilk test and the residuals were normally distributed (*p* < 0.05). Data on hormonal and antioxidant indices were analysed using the repeated measures model ANOVA. When effects of treatment, time, or the interaction between treatment and time were detected (*p* < 0.05), these data were analysed using Fisher post hoc test. The effects of treatment, time, and the interaction between treatment and time are shown under each graph (Figure 1 and Figure 2). Significant differences between groups and differences along time points within each group are indicated by *p* values (Figure 1 and Figure 2) and described in the text. Results were considered significant when *p* ≤ 0.05, and trends were noted when *p* > 0.05 and *p* ≤ 0.01.

## 3. Results

As shown in Figure 1, no differences were observed for all hormone levels measured on day 25 before the expected parturition. The addition of CNT had significant effects on T3 and T4 levels determined on day 5 before calving, resulting in higher levels in the CNT group than in the CON group. Serum estradiol and progesterone levels were comparable between CNT and CON cows on day 5 before calving. Serum levels of estradiol increased while progesterone decreased in CNT and CON cows between days 25 and 5 before calving.

No differences were observed in the antioxidant indices and metabolites measured on day 25 before the expected calving. On day 5 before calving, serum activities of T-AOC (FRAP), GPx, GSH, and SOD were higher, whereas serum levels of TBARS, LDH, and AST were lower in the CNT group than in the CON group (Figure 2). GPx, GSH, SOD, and LDH increased in CNT and CON cows between days 25 and 5 before calving. T-AOC concentration increased in CNT cows, while AST activities increased in CON cows between days 25 and 5 before calving.

## 4. Discussion

The main objective of this study was to determine the factors contributing to improved antioxidant status in CNT cows by determining the concentrations of thyroid and sex hormones of close-up cows. The close positive correlation between improved thyroid status and oxidative defence mechanisms has been previously demonstrated in humans and laboratory rodents [33]. Moreover, oxidative stress has also been suggested to be associated with thyroid hormone dysfunction [19]. In a human study, Baskol et al. [34] showed significantly increased malondialdehyde (MDA) levels and a negative correlation with T3 and T4 levels and decreased PON 1 activity in subjects with primary hypothyroidism. Limited data on thyroid function and antioxidant system during the periparturient period are available for large ruminants, especially dairy cows. Weitzel et al. [22] reported a decline in T3 and T4 in response to heat stress in the prepartum period, which was associated with decreased oxidative defence in early lactation. In our study, we used a model of dietary supplementation with chestnut tannins to gain information on antioxidant indices such as SOD, GPx, and GSH and thyroid hormone levels in prepartum dairy cows. Previous studies defined the antioxidant and free radical scavenging potential of chestnut tannins [7,8,9]. Our results show that CNT cows have higher activities of SOD and GPx as well as total antioxidants and lower TBARS levels than CON cows, indicating the ability of the antioxidant defence system and/or lower level of oxidative stress in CNT cows. The mechanism of action by which dietary tannins exert an antioxidant effect on ruminants is not fully understood [35]. As previously mentioned, rumen metabolism and the bioavailability of tannins differ in relation to the source and form of tannins, thus it seems that hydrolysable tannins reduce proteolysis and methane production more by inhibiting functional rumen microbes, while condensed tannins act more by reducing protein and carbohydrate degradation with their molecules binding capacity [15]. Generally, hydrolysable tannins can be catabolized to acetic and butyric acids with 3-hydroxy-5-oxohexanoate pathways in the rumen [36]. On the other hand, condensed tannins are difficult to be degraded in the rumen and have a greater affinity to carbohydrates than hydrolysable tannins [37]. Liu et al. [7] hypothesized that chestnut tannins display selectivity in inducing antioxidative enzymes gene expression (SOD, GPx) most likely via activation of nuclear factor E2-related factor 2 (Nrf2) [38]. In addition, the antioxidant effects of polyphenols could be a result of their redox properties: quenching singlet and triplet oxygen, decomposing peroxides, or adsorbing and neutralizing free radicals [39].

Analysis of thyroid hormones in our study showed for the first time that cows supplemented with chestnut tannins had significantly higher T3 and T4 levels than cows in the control group on day 5 before parturition. As confirmed in previous studies, low thyroid hormone concentrations in prepartum dairy cows have been linked with the development of metabolic diseases postpartum, thus this effect of chestnut tannins can be defined as beneficial for cows and provides good dietary practice for metabolic challenges in the postpartum period [24]. Similar results were reported by Al-Amoudi [40] who investigated the thyroid effects of ginger extracts, which also contain tannins as one of the bioactive compounds. He reported that ginger administration improved the histopathological changes in thyroid glands and increased the T3 and T4 levels in rats treated with lambda-cyhalothrin. Several mechanisms could be responsible for this improved thyroid function in dry dairy cows supplemented with chestnut tannins. As previously reported, and also in our experiment, tannin supplementation had no effect on sex hormone concentrations compared to cows that were not supplemented [41]. Therefore, 17β-estradiol and progesterone are probably not involved in the differences in serum T3 and T4 concentrations between groups after tannin supplementation. The higher prepartum T3 and T4 concentrations in response to plant extract supplementation in the present and previous studies may be the result of a direct antioxidant effect on the thyroid gland leading to less oxidative damage to thyroid tissue [40,42]. Abliz et al. [43] demonstrated that NADPH oxidase may be involved in the pathophysiology of thyroid dysfunction by regulating thyroid oxidative stress. It has been demonstrated that the tissue damage caused by peroxide radicals can be influenced by tannins in part by reducing the activity of NADPH oxidase [44,45]. Therefore, tannins acting on NADPH oxidase directly or by scavenging free radicals could promote the synthesis of thyroid hormones. In addition, the enhancement of antioxidant capacity by the polyphenol mixtures of CNT extracts could be a contributing factor to the higher thyroid hormone levels in CNT cows. Accordingly, chestnut tannins might contribute to the increased T3 levels in the blood of CNT-fed cows by protecting 5’-monodeiodinase from free radicals and/or by promoting extrathyroidal conversion of T4 into T3, which is at least partially caused by GSH [46]. Finally, hormone levels are regulated not only by synthesis and secretion but also by degradation and excretion [47]. There is a study demonstrating the inhibitory effect of tannic acid on these processes in the liver of rats and humans [48]. Therefore, the effect of chestnut tannins on blood thyroid hormone concentration may be related to the decrease in the clearance of T3 and T4.

Thyroid hormones have been suggested to interfere with the antioxidant system by activating antioxidant defences and reducing lipid peroxidation [21]. In the present study, the stimulatory effects of chestnut tannins on antioxidant capacity are partly due to the activation of endogenous antioxidant defences against free radicals. These results are consistent with the observation of Liu et al. [7], who found that the activities of SOD and GPx were increased by chestnut tannins in the liver and plasma of transition dairy cows. The positive correlation between T3 and SOD and PON 1 is often reported in hypothyroidism because of the important role of thyroid hormones in lipid metabolism and the antioxidant function of these enzymes [18,33]. Similarly, Baskol et al. [34] reported the therapeutic benefit of levothyroxine in increasing SOD and PON 1 serum levels in hypothyroidism, and research has also shown that T3 administration leads to transcriptional upregulation of enzymatic antioxidants such as superoxide dismutase and catalase in hypothyroid rats [49]. We found that serum T3 and T4 levels in cows supplemented with chestnut tannins were higher than those in the control group, suggesting that thyroid hormones may play an important role in improving SOD in CNT cows. Extracellular SOD (EcSOD) is an enzyme found predominantly in extracellular fluids such as plasma, lymph, and synovial fluid, and is the only known antioxidant enzyme that specifically scavenges O_2_• in the extracellular space [50]. However, its redox-modulating action is not limited to controlling the concentration of this radical. By suppressing the accumulation of superoxide, EcSOD also prevents the spontaneous dismutation of superoxide to H_2_O_2_. In this way, EcSOD can also inhibit the production of •OH via the Fenton and Harbor–Weiss reactions that directly damage the cell membrane [51]. Thus, the increased activity of this enzyme in the group of cows treated with chestnut tannins on day 5 before parturition suggests a protective effect of EcSOD against oxidative stress and cellular damage.

ROS-induced oxidative damage also provokes lipid peroxidation. Lipid peroxidation products can be shifted from the site of their formation to other sites in the organism, thus transmitting information about the origin of the primary oxidative damage [52]. The extent of lipid peroxidation in the cows studied was monitored using TBARS values including MDA, one of the end products of the cyclic reaction of lipid peroxidation. According to our TBARS results, lipid peroxidation was shown to be lower in the presence of the supplied tannins. TBARS are widely used indicators of oxidative stress in dairy cows as well [52], and the observed lower TBARS in CNT cows may be related to free radical scavenging and/or improved SOD [7,52]. Furthermore, in order to indirectly monitor the level of lipid peroxidation, we also evaluated the release of the enzyme LDH from cells. Since the increased extracellular activities of AST and LDH could be caused by impaired cell membrane integrity that occurs in liver damage under the influence of free radicals, the lower AST and LDH activities in CNT cows in our study support the potent antioxidant properties of chestnut tannins [53]. The consistency of the results of the different parameters (TBARS, AST, and LDH) for the assessment of lipid peroxidation confirmed their reliability in determining membrane damage, demonstrating the beneficial roles of CNT in attenuating oxidative damage near parturition. Chestnut tannin supplementation in our study improved antioxidant protection 5 days before calving, activating the antioxidant defence enzymes GPx and SOD, preventing redox processes, reducing the degree of cell membrane damage, and releasing intracellular enzymes from hepatocytes. By activating mechanisms of antioxidant protection and neutralizing ROS, tannins might protect thyroid tissue from damage, thus allowing for higher T3 and T4 synthesis.

Although our study provides a link between the antioxidative defence mechanism and thyroid function in the CNT group, further research is needed to clarify all the factors involved in these processes and their interaction. Hepatic deiodinase activity is an important control point for the regulation of thyroid status under various physiological and pathological conditions [54]. Therefore, it would be interesting to investigate the effect of chestnut tannins on hepatic deiodinase gene expression to further dissect the cause of improved thyroid status in prepartum cows. Furthermore, since our results show increased GPx activity in the CNT group, it would be interesting to investigate selenium concentration in cattle supplemented with chestnut tannins. Indeed, selenium is an important cofactor for both GPx and deiodinase enzymes [55].

## 5. Conclusions

The use of chestnut tannins in close-up dry cows has the potential to improve thyroid function near parturition. Therefore, dietary supplementation with chestnut tannins is important not only for improving antioxidant status but also for achieving proper thyroid function in dairy cows. Further studies on the implications of such a dietary approach in formulating close-up period rations will be needed, especially in specific conditions of altered thyroid function (e.g., heat stress).

## Figures and Tables

**Figure 1 metabolites-13-00334-f001:**
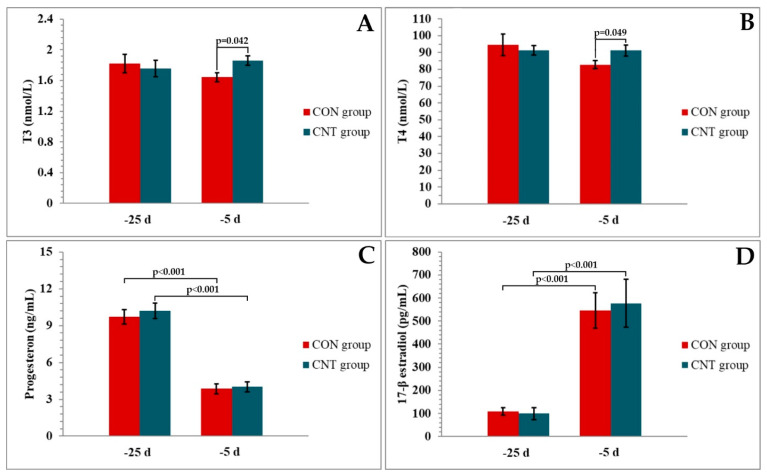
Concentrations of (**A**) triiodothyronine (T_3_), (**B**) thyroxine (T_4_), (**C**) progesterone, and (**D**) 17-β estradiol in serum of CON and CNT groups of cows at 25 and 5 days before expected parturition. Values are expressed as mean ± SE. (**A**) Effects of treatment (*p* = 0.332), time (*p* = 0.754), and interaction between treatment and time (*p* = 0.050); (**B**) effects of treatment (*p* = 0.532), time (*p* = 0.151), and interaction between treatment and time (*p* = 0.050); (**C**) effects of treatment (*p* = 0.742), time (*p* < 0.001) and interaction between treatment and time (*p* = 0.753); (**D**) effects of treatment (*p* = 0.540), time (*p* < 0.001), and interaction between treatment and time (*p* = 0.531). Significant differences between groups as well as differences along the time points within each group are indicated by *p* values (Fisher post hoc test). −25 d = first day of chestnut tannins supplementation (25 ± 2 days before parturition). −5 d = 5 days before parturition.

**Figure 2 metabolites-13-00334-f002:**
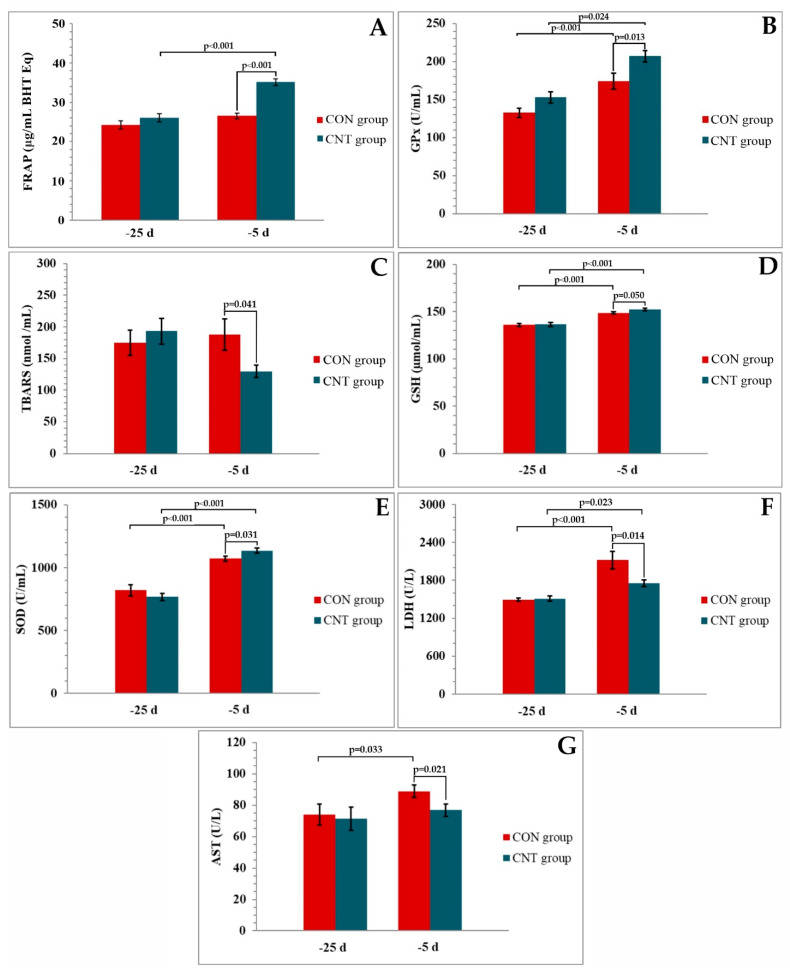
Activities of (**A**) total antioxidants (FRAP), (**B**) glutathione peroxidase (GPx), (**E**) superoxide dismutase (SOD), (**F**) lactate dehydrogenase (LDH), and (**G**) aspartate aminotranspherase (AST), and concentrations of (**C**) thiobarbituric acid reactive substances (TBARS) and (**D**) reduced glutathione (GSH) in serum of CON and CNT groups of cows at 25 and 5 days before expected parturition. Values are expressed as mean ± SE. (**A**) Effects of treatment (*p* < 0.001), time (*p* < 0.001), and interaction between treatment and time (*p* < 0.001); (**B**) effects of treatment (*p* < 0.01), time (*p* < 0.001), and interaction between treatment and time (*p* = 0.687); (**C**) effects of treatment (*p* = 0.100), time (*p* = 0.300), and interaction between treatment and time (*p* = 0.221); (**D**) effects of treatment (*p* = 0.271), time (*p* < 0.001), and interaction between treatment and time (*p* = 0.291); (**E**) effects of treatment (*p* = 0.613), time (*p* < 0.001), and interaction between treatment and time (*p* = 0.020); (**F**) effects of treatment (*p* = 0.121), time (*p* < 0.001), and interaction between treatment and time (*p* = 0.043); (**G**) effects of treatment (*p* = 0.064), time (*p* = 0.101), and interaction between treatment and time (*p* = 0.133). Significant differences between groups as well as differences along the time points within each group are indicated with *p*-values (Fisher post hoc test). −25 d = first day of chestnut tannins supplementation (25 ± 2 days before parturition). −5 d = 5 days before parturition.

**Table 1 metabolites-13-00334-t001:** Ingredients and chemical composition of the far-off and close-up diets.

	Diet	
Item	Far-Off	Close-Up
Ingredient, g/kg of DM		
Alfalfa hay	173	170
Corn silage	374	369
Alfalfa haylage	115.3	91.3
Molasses	-	38.3
Corn grain	36.5	113
Barley	59.7	29.4
Soyabean cake (42%CP)	-	15.7
Soyabean meal (44%CP)	-	1.96
Sunflower meal (34%CP)	17.4	85.4
Wheat bran	153	64.8
Calcium carbonate	60.5	4.90
Monocalcium phosphate	2.49	1.96
NaCl	1.66	4.90
Sodium bicarbonate	-	0.98
Vitamine Mineral Mix	5.80	7.85
DMI (kg/day)	12.1	10.2
Chemical composition		
Energy		
NE_L_(Mcal/kg of DM)	1.39	1.57
CP (g/kg of DM)	122	140
RDP (g/kg of DM)	92	110
RUP (g/kg of DM)	30	30
MP (g/kg of DM)	76.4	81.8
NDF (g/kg of DM)	477	382
ADF (g/kg of DM)	305	236
NFC (g/kg of DM)	335	407
Ether Extract (g/kg of DM)	21	25
Ca (g/kg of DM)	7	7
*p* (g/kg of DM)	4	4
Ash (g/kg of DM)	112	82.5

DM = dry matter; CP = crude protein; DMI = dry matter intake; NEL = net energy for lactation; RDP = rumen degradable protein; RUP = rumen undegradable protein; MP = metabolisable protein; NDF = neutral detergent fibre; ADF = acid detergent fibre; NFC = non-fibre carbohydrate.

## Data Availability

The data presented in this study are available on request from the corresponding author (dani@vet.bg.ac.rs) and are available under code at the https://mitimetcattle.vet.bg.ac.rs/ accessed on 1 September 2022. The data are not publicly available due to the current requirements of the project that supported this research.

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
