# Peer review of "Dietary Supplementation of Chestnut Tannins in Prepartum Dairy Cows Improves Antioxidant Defense Mechanisms Interacting with Thyroid Status"

_metabolites, 2023, doi:10.3390/metabo13030334_

Round 1

Reviewer 1 Report (Previous Reviewer 5)

Manuscript ID: metabolites-2193342

The revised manuscript contributes original metabolic research. It is quite well-written, clearer, relevant for the ruminant metabolism and presented in a well-structured manner. The authors have considered all the comments and suggestions and made a good response to the revised manuscript. Clearer details have been made in research methodology, discussion, limitations and implementations. No further comments.

Author Response

We thank the reviewer for his time and positive evaluation of our manuscript.

Reviewer 2 Report (New Reviewer)

Comments and Suggestions for Authors of the article draft “Dietary supplementation of chestnut tannins in prepartum dairy cows improves antioxidant defense mechanisms inter-acting with thyroid status”.

The draft is a good document, well written, with a good review of the literature, and which addresses a current and interesting topic related to stress in dairy cows.

I only have two observations about the draft.

Line 185 states that " trends were noted if 0.05<p<0.1.”, However, I don't know if this expression is correct, since logic indicates that it should be p>0.5<0.1. Please check that expression.

In lines 208 and 234 (Figures 1 and 2), the symbol (green box) that identifies the CNT group is spliced with the text. I suggest separating them.

Author Response

The draft is a good document, well written, with a good review of the literature, and which addresses a current and interesting topic related to stress in dairy cows.

ANSWER

We thank the reviewer for his time and positive evaluation of our manuscript

I only have two observations about the draft.

Line 185 states that " trends were noted if 0.05<p<0.1.”, However, I don't know if this expression is correct, since logic indicates that it should be p>0.5<0.1. Please check that expression.

ANSWER

In accordance with the recommendation of reviewer  we check and correct this part of sentence.

Please see revised verson of the manuscript:

„Results were considered significant when p≤0.05, and trends were noted when p > 0.05 and p ≤ 0.01.“

Instead of

“Results were considered significant if p≤0.05 and trends were noted if 0.05<p<0.1.”

In lines 208 and 234 (Figures 1 and 2), the symbol (green box) that identifies the CNT group is spliced with the text. I suggest separating them.

ANSWER

In accordance with the recommendation of reviewer we check and correct this typo error, please see lines 221; 246 of revised version of the manuscript.

Reviewer 3 Report (New Reviewer)

This is a report of feeding chestnut tannins to dairy cows for the last 25 days of pregnancy. The tannins increased thyroid hormones at 5 days before calving and decreased indicators of oxidative stress.

The Introduction covers the topic, with appropriate references to explain the background and reasons for this particular work. The study appears to be well-conducted with measurements recorded before commencement of tannin feeding and again shortly before calving, a critical period for dairy cows.

The results are well presented, showing differences between tannin supplemented cows and the control group before commencement and before calving for a range of relevant measures of thyroid hormones, sex hormones, oxidative function  and tissue enzymes.

The Discussion explains the results in the light of other studies and considers the implications of this work and potential future research.

In the Conclusions there is possibly an over-emphasis on chestnut tannins specifically, whereas the Introduction suggests that other tannins might be equally suitable, although there may be a difference in effect between condensed tannins and hydrolysable tannins. However, since only one type of tannins was used here, it may be difficult to suggest other suitable tannins for further work.

Author Response

This is a report of feeding chestnut tannins to dairy cows for the last 25 days of pregnancy. The tannins increased thyroid hormones at 5 days before calving and decreased indicators of oxidative stress.

The Introduction covers the topic, with appropriate references to explain the background and reasons for this particular work. The study appears to be well-conducted with measurements recorded before commencement of tannin feeding and again shortly before calving, a critical period for dairy cows.

The results are well presented, showing differences between tannin supplemented cows and the control group before commencement and before calving for a range of relevant measures of thyroid hormones, sex hormones, oxidative function  and tissue enzymes.

The Discussion explains the results in the light of other studies and considers the implications of this work and potential future research.

In the Conclusions there is possibly an over-emphasis on chestnut tannins specifically, whereas the Introduction suggests that other tannins might be equally suitable, although there may be a difference in effect between condensed tannins and hydrolysable tannins. However, since only one type of tannins was used here, it may be difficult to suggest other suitable tannins for further work.

ANSWER

We thank the reviewer for his time and positive evaluation of our manuscript. Bearing in mind that we used only chestnut tannins, we have to specify this in the conclusion. However, according to reviewer suggestion we rephrase the part of the introduction section so clearly define differences in types and sources of tanins:

Please see revised verson of the manuscript:

"The ability of tannins to improve feed efficiency and antioxidant status in dairy cows has been well known and they have been often supplemented in the diet of dairy cows [8–10]. Tannins have also been suggested as effective alternatives to antibiotics to suppress rumen methanogenesis through their antimicrobial activity [11,12]. Extracts of both, condensed and hydrolysable tannins, can be incorporated into dairy cows’ diet, whereas the quality and quantity of their effects may differ depending on tannin source [8,9,13]. The use of condensed tannins in ruminant diet due to its protein precipitation, anti-microbial and anti-parasitic activities has been widely researched [13–15]. However, compared to condensed tannins, hydrolysable tannins have a greater biological activity especially in regards to their antioxidant properties. [9,16]. Hydrolysable tannins from sweet chestnuts are effective at scavenging ROS and at protecting liver and kidney tissues against oxidants as well as liposomes from lipid peroxidation [8,9]."  

Instead of

“The ability of tannins to affect several aspects of ruminant nutrition and to improve antioxidant status in dairy cows has been well known and they have been often supplemented in the diet of dairy cows. The diets of dairy cows could be supplemented with condensed or hydrolysable tannins where the dietary effect may differ depending on the tannin source [8–10]. Condensed tannins are widely supplemented in diet due to their low prices. Whereas, compared to condensed tannins, hydrolysable tannins have a greater biological activity especially in regards to their antioxidant properties [9,11]. Hydrolysable tannins from sweet chestnuts are effective at scavenging ROS (O2-) and at protecting some tissues against oxidants as well as liposomes from lipid peroxidation [8,9]. “

Reviewer 4 Report (New Reviewer)

The manuscript "Dietary supplementation of chestnut tannins in prepartum dairy cows improves antioxidant defense mechanisms interacting with thyroid status" by Prodanović et al. presents data on how dietary chestnut tannin extract affects the activities of several antioxidant enzymes and the concentration of thyroid and sex hormones in plasma of prepartum dairy cows. The results are generally interesting but there are some major concerns:

In the manuscript, it is mentioned several times that this study seeks to identify a mechanistic relationship of thyroid function and antioxidant status. However, the study design does not allow for this because only plasma concentrations/enzyme activities are presented and thus, the respective sections throughout the manuscript must be adjusted. Authors could analyse correlations to show some more relationships but again, these would not be causal.

The discussion section is in many parts not precise, and it remains partially unclear what authors want to express (see detailed comments below).

The manuscript contains many grammar errors. I pointed out a few of those below but for completeness, this manuscript must be revised by a native speaker or a professional English language editing service.

Please find my detailed comments below:

LL17/18: Please add a comment about why investigating chestnut tannins would be relevant. Why are chestnut tannins used in cow diets?

L23: "on first day", that means before the first time supplementing tannins, I assume? Please specify.

L27: The sentence reads "activities of … capacity". The term activities refers to the enzymes but not to T-AOC. Please rephrase the sentence.

L28: Reduced glutathione … Something is missing here, because it is not the activity GSH that is measured. Concentration?

L30: …transferase… (instead of …transpherase…); Also, were these really "levels" that were measured or again activities?

L31: differed (past tense). But actually, it should then probably read "did not differ between…"?

L32/33: I do not agree with this statement. Just because several parameters were affected in addition to the thyroid hormones does not prove a causal connection. Tannins could affect hormones and e.g. antioxidant enzymes via different mechanisms. Please remove/replace this sentence.

LL46–48: This is not true for all cell membranes. Please indicate which organ is damaged when AST and LDH activities are increased in plasma.

L48: What do you mean by "the final effect of ROS"? Please clarify.

L49: It says "periods", so plural. Which other periods are you referring to in addition to the peripartum period?

L52: Please specify what you mean by "several aspects of ruminant nutrition" and provide references.

LL54–56: This sentence mixes two different factors: the type of tannin (condensed v s hydrolysable) and the tannin source. They are partially interdependent which is not evident from this sentence. Please clarify.

L56: I don't think that the price is the main reason for the supplementation of condensed tannins but rather their biological effects. These should be mentioned at some point (see comment for abstract above).

LL57–58: This sentence is incomplete (grammar).

L59: O2- is not a correct formula. I would recommend to just delete this example.

LL59/60: Please specify "some tissues".

LL73/74: Why is this "in addition"? The previous sentence already states that thyroid hormones may regulated antioxidant defense systems. Or would this now be via another mechanism? Please clarify.

L80: This should probably be "thyroid hormones" instead of "thyroid"?

L81: Which free hormone levels? Thyroid again, I think?

LL103–105: Please provide the amounts of hydrolysable and condensed tannins that were provided via the chestnut product.

LL106–108: Please provide the diet composition. Were there other tannin sources/what was the daily intake of tannins from the TMR?

LL116/117: Even though the data of feed intake is not shown, please provide at least average feed intakes per group here in the text.

L121: One "day" too much.

L148: Amount for deionized water is missing.

L176: It is not the normality of data distribution that should be tested but the one of the residuals. Please confirm that normality was correctly tested.

L190 (and the rest of the text): Do not repeat p-values from figures in text and vice versa.

LL215–217: The increase of T-AOC by tannin supplementation was already described in L211.

L237: The study did not investigate any mechanisms. Please rephrase.

L239: Please the direction of interaction. Is this actually a correlation? Positive or negative?

LL248–250: It is not clear why authors make reference here to a difference between pre- and postpartum cows. The cited studies as well as the experiment presented in this manuscript were performed in pre-partum cows. What is the point of referring to postpartum cows here? Please clarify.

LL250–252: I do no agree that the study design allows for investigating a causal relationship of antioxidant enzyme activities and thyroid hormones. The study design allows for describing effects of tannin supplementation on antioxidant enzymes and thyroid hormones. They may be affecting via different mechanisms. If any, then correlations could be calculated which, again, do not allow for causal interpretation. Please rephrase accordingly.

LL252–256: It is important to distinguish the underlying mechanisms for antioxidant effects of tannins. Polyphenols acting as free radical scavengers do not necessarily induce gene expression or enhance activity of antioxidant enzymes. It would probably help to better describe the mechanisms via which tannins may increase antioxidant enzyme activity. Also, authors should include a section on the rumen stability of hydrolysable tannins. It would also be interesting to add data on the plasma polyphenol concentration, if there is sample material left and the authors could analyse this parameter.

LL258/259: Can these higher T3 and T4 concentrations be regarded as positive? Why (not)? I am missing (potentially already in the introduction) a brief summary of what thyroid hormones do and why they are important with particular focus on the prepartum period.

L260: I would be careful calling this a stimulation of the thyroid because there was no increase from -25 d to -5 d in the CNT group. It could also be avoiding an inhibition that may occur otherwise few days prepartum, but this was also not significant. Sentence needs to be rephrased.

LL269–280: How are “Second, …decreased oxidative damage…” and “Third, improvement in antioxidant capacity…” different? To my understanding, these describe the same potential underlying effects. Please combine or rephrase to make the difference clearer.

LL285/286: This sentence is a contradiction. Thyroid stimulation is not the same as decrease in clearance. These are mechanisms that can occur completely independent from one another.

L293: Have authors calculated the correlation between T3 and SOD in the present study? What is the result? This might be worth adding.

L323: AST is not necessarily a marker for lipid peroxidation but for liver damage. Please add this information.

LL328–330: This is one possibility, but the previous discussion also states that the inverse mechanism is possible. By improving SOD activity, tannins might protect thyroid tissue from damage, thus allowing for higher T3 and T4 formation. The results of the present study do not allow for identifying what comes first. Please rephrase accordingly.

L331: The study does not provide insights into a more detailed mechanism of action, as lined out in my previous comments. Please rephrase.

All figures: In general, please always provide three decimal places for the p-values.

Figure 1:

-          A, y-axis: Decimal point should be a point, not a comma.

-          D: The y axis does not start at 0 and the upper scale bar of the y-axis does not have a number. The latter is also true for A and C in the same figure. Please adjust.

-          Figure caption: Delete the listing of individual p-values. The figure is showing the significant p-values already, so this is an unnecessary repetition. The legend is already visible in each figure and does not need to be repeated in the caption.

Figure 2:

-          F and G: The upper scale bar of the y-axis does not have a number.

-          Figure caption: Delete the listing of individual p-values. The figure is showing the significant p-values already, so this is an unnecessary repetition. The legend is already visible in each figure and does not need to be repeated in the caption.

Round 2

Reviewer 4 Report (New Reviewer)

The manuscript "Dietary supplementation of chestnut tannins in prepartum dairy cows improves antioxidant defense mechanisms interacting with thyroid status" by Prodanović et al. has greatly improved since the previous version. Some of my comments regarding the figures were not addressed:

-          In my last review, I had mentioned that P values should always show three decimal places. In all figures, the P values depicted in the figure still list only two decimal places. This must be changed (not the figure caption but the figure itself).

-          The figure captions still show the legend, which is also shown in each individual figure. Please delete legend from both figure captions.

There are a few further remarks:

 L29: Please change to “…with higher serum total antioxidant capacity…” (delete “concentration of”).

 L65: Remove “.” following “properties”.

 L81: Please specify the types of tissues instead of “some tissues”.

 LL112/113: Following the request to add information about the type of tannin used the experiment, the authors added a reference to a paper describing tannin analysis. The paper is not describing the analysis of the exact product (Tanimil SCC) used in the present study, so I am afraid this is not a valid reference. Tannin composition of natural products varies e.g., according to growing conditions. If not analysed in the course of the study, at least the producer should be able to provide information about %CT and %HT. If this information is not available, please clearly state that in the manuscript.

 LL125/126: The number of decimal places should be the same for both groups. Please indicate if this is mean±SD or mean±SEM. SD should have same number of decimal places as the mean. SEM should have one more decimal place than mean.

 LL188-198: Authors have removed the statement about normality testing of data/residuals. This is not acceptable. It must be indicated that residuals were tested for normal distribution and that data used in the repeated measures ANOVA fulfilled the respective assumptions.

 L267: “The mechanism of action” instead of “The action mechanism”, “exert” instead of “exerts”.

 L269: “differ” instead of “differs”.

 L271: “act” instead of “acts”.

 L272: “their” instead of “its”.

 L274: “condensed tannins are” instead of “condensed tannins is”.

 L275: “and have” instead of “and has”.

 L355: “improved” instead of “improves”.

 L358: Start sentence with “By…” (By activating mechanisms of…)

 L360: Insert “a” before “link”.

 L361: “to clarify” instead if “to clarified”.

 L362: Insert “and their interaction” following “in these processes”.

 Table 1: For values with three digits before the comma (“.”), please remove the decimal places. For values with two digits before the comma, only one decimal place should be given. When items were not included in one of the diets, please indicate by placing “-“ or “0” in the table. Provide one more decimal place for NEL in the far-off diet.

Author Response

This manuscript is a resubmission of an earlier submission. The following is a list of the peer review reports and author responses from that submission.

Round 1

Reviewer 1 Report

The article is an interesting study of dietary supplementation with chestnut tannins in prepartum dairy cows to improve antioxidant defense mechanisms in interaction with thyroid status. The results are well presented and support the discussion. This study is part of a previously published paper (reference 9) that found that supplementation of the diet with chestnut tannins during lactation had no adverse effects on prepartum metabolic profiles. Instead, this feeding regimen had a more beneficial effect on antioxidant capacity and colostrum quality than feeding the control diet
There is one minor comment: line 107 - Explain the DMI.

Author Response

There is one minor comment: line 107 - Explain the DMI.

Answer: We provide the full name of abbreviation in the revised version of the manuscript (please see line 146 in the revised version of the manuscript).

Reviewer 2 Report

Dear authors, it seems to me that this manuscript has great relevance in the scientific world. However, many points affect the quality of the manuscript.

General comments:

Abstract: The abstract is not complete. There is a lack of information of your results. Remember, the abstract is the document that first represents your manuscript. The conclusion does not support the objective. E.g.: T4 concentration increased by 5% compared to the control diet.

Results: Improve the writing style. You need to show more in the results topic. E.g.: T4 concentration increased by 5% when cows were fed diets containing 1.96 g of CNT tannins.

Specific comments:

Line 31: What do AST and LDH mean? Describe the full name the first time it appears, both in the abstract and in the body of the manuscript.

Lines 33-34: Speculative idea.

Lines 39-40: Add citation.

Line 57: You need to cite those several studies. At least three recent studies.

Lines 57-59: Intricate text. Please divide this text into two ideas.

Line 59-60: Incomplete idea. I know that the following ideas complete this text; however, this idea needs a few words to complete and not seem like an isolated idea.

Line 63: Is the focus oxidative stress or energy expenditure? If there is a correlation between these variables, describe them in the text.

Line 77-78: According to?

Lines 78-80: According to?

Lines 84-88: This idea can be placed with the earlier ideas about the thyroid gland and oxidative stress, on lines 66-76. In this part it is repetitive.

Line 88: Add the hypothesis before the objective.

Line 89: Thyroid what? Thyroid metabolism; thyroid function; thyroid health; ….

Line 89: sex hormones? Try to improve this part, because when you read it, it implies that the study is a comparison between males and females.

Lines 92-94: Is there a protocol number?

Figure 1, 2: Add the P-values to the figure.

Line 196: Similar phrases are described in the introduction and in the materials and methods topics. So please change "The" instead of "This study extends our previous work, which showed that".

Lines 198-199: Ok, this is the differential of your work. However, what is the point? There is already knowledge about the effects of tannins as antioxidants, how are chestnut tannins different? Greater concentration? higher solubility? Greater hydrolyzable capacity? Greater indigestibility in the rumen that allows greater availability to be absorbed in the intestine? Those points should be described primarily in the introductory topic and here as an option.

Lines 199-201: Similar to the comment on line 196. For me these lines do not add relevant information to improve your manuscript. I recommend removing them.

Lines 205-207: Are you sure? In a quick search (5 minutes), I found three manuscripts on this topic:

-Schering, L., Albrecht, E., Komolka, K., Kühn, C., & Maak, S. (2017). Increased expression of thyroid hormone responsive protein (THRSP) is the result but not the cause of higher intramuscular fat content in cattle. International journal of biological sciences, 13(5), 532.

- Dehghan Shahreza, F., Seifi, H. A., & Mohri, M. (2022). The relationship between body condition score, thyroxin, and health condition and serum energy indices, insulin like growth factor-1, and lipids profile over the transition period in Holstein dairy cows.

- Todini, L. (2007). Thyroid hormones in small ruminants: effects of endogenous, environmental and nutritional factors. Animal, 1(7), 997-1008.

Line 209: If this is the main objective, why did you describe the full text before?

Lines 196-212: I recommend rewriting this part into a more concise and specific part that correlates with your study and not past studies.

Line 217: According to the instructions for authors, the correct citation is: Baskol et al. [28]. Correct here and throughout the text.

Line 235: Here you focus obviously on the chestnut tannin. For this reason, as previously mentioned, how is chestnut tannin different so that its results are so relevant? Phrase to think about: tannin is tannin independent of the source.

Line 238: Change “They” instead of “He”.

Author Response

Dear authors, it seems to me that this manuscript has great relevance in the scientific world. However, many points affect the quality of the manuscript.

General comments:

Abstract: The abstract is not complete. There is a lack of information of your results. Remember, the abstract is the document that first represents your manuscript. The conclusion does not support the objective. E.g.: T4 concentration increased by 5% compared to the control diet.

Answer: We are thankful for this remark. We try to improve the abstract in accordance with this suggestion and beside the P values for all obtained parameters we also provide the differences in percentage values (please see lines 28-34 in the revised version of the manuscript). Bearing in mind that the differences range from 2.32 % for GSH to 32.41% for T-AOC and that for thyroid hormones were 13.21 % and 10.25 % for T3 and T4, respectively, we believe that we can stand behind our conclusions with relative certainty. Starting from the fact known in the previous researches that CNT supplementation leads to improvement of antioxidant protection mechanisms in dairy cows, the main objective of our research was to examine the impact of CNT supplementation on thyroid status which was previously connected in antioxidative defence mechanisms in various physiological and pathological states. In the revised version of the apstract conclusion (please see lines 37-40 in the revised version of the manuscript), we give a direct answer to the question from the objective, but also defined the direction of future research. A prerequisite for the use of CNT as a supplement is a complete knowledge of the mechanisms of its action in order to perform the supplementation in the right way, the appropriate dosege and at the right time in production cycle in dairy cows.

Results: Improve the writing style. You need to show more in the results topic. E.g.: T4 concentration

Answer:  In accordance with one of the previous answers, we changed the results section and beside the P values for all obtained parameters we also provide the differences in percentage values (please see lines 28-34 in the revised version of the manuscript)

Specific comments:

Line 31: What do AST and LDH mean? Describe the full name the first time it appears, both in the abstract and in the body of the manuscript.

Answer: We provide the full name of abbreviation in the revised version of the manuscript (please see line 33 in the revised version of the manuscript).

Lines 33-34: Speculative idea.

Please see the responses to the comments above. We changed the conclusion in accordance with the recommendation of the reviewer (please see lines  37-40 in the revised version of the manuscript).

Lines 39-40: Add citation.

Answer:  We provide the citation according to  the reviewer recommendation (please see line 48 in the revised version of the manuscript).

Line 57: You need to cite those several studies. At least three recent studies.

Answer:  In accordance with the further recommendations of the reviewer, this part of the introduction was excluded from the revised version of the manuscript.

Lines 57-59: Intricate text. Please divide this text into two ideas.

Answer:  In accordance with the reviewers recommendations the entire introduction section is revised (please see lines  46-120 in the revised version of the manuscript). The introduction has been rearranged to state the importance of oxidative stress in dairy cows at first, then to analyze the current scientific knowledge related to the influence of tannin supplemntation on oxidative stress, and in final to present a thesis on the possible connection of antioxidant defence mechanisms with thyroid status and sex hormones in CNT supplemented cows. We hope that this rearranged introduction contributed to a better understanding of the text and satisfied the requirements of the reviewer.

Line 59-60: Incomplete idea. I know that the following ideas complete this text; however, this idea needs a few words to complete and not seem like an isolated idea.

Answer:  Bearing in mind that we performed a detailed revision of the introduction section, we tried to more clearly connect the interaction of thyroid status and antioxidant defence mechanisms, as well as to point out the possible influence of tannin supplementation on thyroid status, which was the focus of our research (please see lines 80-102 in the revised version of the manuscript).

Line 63: Is the focus oxidative stress or energy expenditure? If there is a correlation between these variables, describe them in the text.

Answer:  We are thankful and agree with the suggestion of the reviewer. In accordance with the suggestion in the introduction, we focused on oxidative stress. So that part of the text would not seem confusing, we removed the part that refers to energy expenditure. please see lines 80-83 in the revised version of the manuscript).

Line 77-78: According to?

Answer:  In accordance with the further recommendations of the reviewer, this part of the introduction was excluded from the revised version of the manuscript.

Lines 78-80: According to?

Answer:  We provide the citation according to  the reviewer recommendation (please see line 79 in the revised version of the manuscript).

Lines 84-88: This idea can be placed with the earlier ideas about the thyroid gland and oxidative stress, on lines 66-76. In this part it is repetitive.

Answer:  We are thankful and agree with the suggestion of the reviewer. In order to avoid repetitions in a line with one of the previous answers, we created a new structure of the introduction. The sentences from the last paragraph (non revised version of the paper) have been transferred to the corresponding paragraphs earlier in the text. please see lines  66-72 in the revised version of the manuscript).

Line 88: Add the hypothesis before the objective.

Answer:  In accordance with this comment  we provide clear hypothesis  before explaining the objective of the study (please see lines 115-117 in the revised version of the manuscript).

Line 89: Thyroid what? Thyroid metabolism; thyroid function; thyroid health; ….

Answer:  We apologize for the oversight. We meant on thyroid function. In the revised version of the paper, the word “function” is added (please see line 119).

Line 89: sex hormones? Try to improve this part, because when you read it, it implies that the study is a comparison between males and females.

Answer: We are thankful and agree with the suggestion of the reviewer. In accordance with the suggestion, we moved the first two sentences about interaction of thyroid function and sex hormones from the discussion section to the introduction (please see lines 94-98 in the revised version of the manuscript).

Lines 92-94: Is there a protocol number?

Answer: We provide a protocol number in the revised version of the manuscript (please see line 123 in the revised version of the manuscript).

Figure 1, 2: Add the P-values to the figure.

Answer: In accordance with the recommendations of the reviewer, P values are added to the graphs for all investigated parameters that showed statistical significance (please see Fig 1 and Fig 2 in the revised version of the manuscript).

Line 196: Similar phrases are described in the introduction and in the materials and methods topics. So please change "The" instead of "This study extends our previous work, which showed that".

Answer: In accordance with the recommendations of the reviewer the discussion has been rearranged and the first paragraph (lines 196-209, non revised version of the paper)  has been excluded from the revised version. In the revised version of the paper in discussion section we immediately focused on the influence of tannin supplementation on thyroid function as well as to explanain  the possible interacions of thyroid function with of antioxidative defence. (please see lines 241-316 in the revised version of the manuscript).

Lines 198-199: Ok, this is the differential of your work. However, what is the point? There is already knowledge about the effects of tannins as antioxidants, how are chestnut tannins different? Greater concentration? higher solubility? Greater hydrolyzable capacity? Greater indigestibility in the rumen that allows greater availability to be absorbed in the intestine? Those points should be described primarily in the introductory topic and here as an option.

Answer: In accordance with the response to the previous comment the first part of the discussion was excluded, and thus this sentence is not transferred to the revised version of the paper. However, at this point we would like to clarify the background of our study and the tannin source we used. Antioxidative properties of tannins are considered to be correlated to a number of hydroxyl groups in their molecule. Chestnut tannins are clasified to a group of hydrolysable tannins with high number of hydroxyl groups due to a significant content of ellagitannins and gallotaninns in chestnut tannin molecule (Smeriglio et al. ,2017).  Additionaly Moccia et al. (2020) demonstrated positive effects of hydrolytic treatment of different tannin sources on their antioxidative properties, with an obvious difference between chestnut (hydrolysable tannins) and quebracho tannins (condensed tannins).  

In order to meet the reviewer requirement in the revised version of manuscript in the introduction section we added the significance of the dietery form and source of tannins which we corroborated with the corresponding reference (please see lines 63-64 in the revised version of the manuscript).

(Smeriglio, A., Barreca, D., Bellocco, E., & Trombetta, D. (2017). Proanthocyanidins and hydrolysable tannins: occurrence, dietary intake and pharmacological effects. British journal of pharmacology, 174(11), 1244-1262 ).

(Moccia, F., Agustin-Salazar, S., Verotta, L., Caneva, E., Giovando, S., D’Errico, G., Panzela, L., d’Ischia, M., Napolitano, A. (2020). Antioxidant properties of agri-food byproducts and specific boosting effects of hydrolytic treatments. Antioxidants, 9(5), 438.)

Lines 199-201: Similar to the comment on line 196. For me these lines do not add relevant information to improve your manuscript. I recommend removing them.

Answer: Please see the answers above, the discussion has been rearranged and the first paragraph (lines 196-209, non revised version of the paper)  has been excluded from the revised version.

Lines 205-207: Are you sure? In a quick search (5 minutes), I found three manuscripts on this topic:

-Schering, L., Albrecht, E., Komolka, K., Kühn, C., & Maak, S. (2017). Increased expression of thyroid hormone responsive protein (THRSP) is the result but not the cause of higher intramuscular fat content in cattle. International journal of biological sciences, 13(5), 532.

- Dehghan Shahreza, F., Seifi, H. A., & Mohri, M. (2022). The relationship between body condition score, thyroxin, and health condition and serum energy indices, insulin like growth factor-1, and lipids profile over the transition period in Holstein dairy cows.

- Todini, L. (2007). Thyroid hormones in small ruminants: effects of endogenous, environmental and nutritional factors. Animal, 1(7), 997-1008.

Answer: We apologize if this sentence caused confusion. Our goal was to point out the lack of studies in cows proving the association of hypothyroidism and  antioxidative defense mechanisms. The link between thyroid hormones and lipid metabolism is certain in both humans and domestic animals. However this part of the discussion is now excluded in accordance with the earlier recommendations of the reviewer.

Line 209: If this is the main objective, why did you describe the full text before?

Answer: Please see the answers above, the discussion has been rearranged and the first paragraph has been excluded from the revised version.

Lines 196-212: I recommend rewriting this part into a more concise and specific part that correlates with your study and not past studies.

Answer: Please see the answers above, the discussion has been rearranged and the first paragraph has been excluded from the revised version.

Line 217: According to the instructions for authors, the correct citation is: Baskol et al. [28]. Correct here and throughout the text.

Answer: We have corrected references and deleted the year of publication in this place and throughout the text (please see lines 265, 269, 285, 301, and 309 of the revised version of the paper).

Line 235: Here you focus obviously on the chestnut tannin. For this reason, as previously mentioned, how is chestnut tannin different so that its results are so relevant? Phrase to think about: tannin is tannin independent of the source.

Answer: Please see answer regards to lines 198-199 comment. We try the explain the significance of the dietery form and source of tannins in the reviewer response and in manuscript text.

Line 238: Change “They” instead of “He”.

Answer: Given that it is  a single author study Al-Amoudi we left the word “He”. Please see the full citation below:

Al-Amoudi, W.M. Toxic Effects of Lambda-Cyhalothrin, on the Rat Thyroid: Involvement of Oxidative Stress and Ameliorative Effect of Ginger Extract. Toxicol. Reports 2018, 5, 728–736, doi:10.1016/J.TOXREP.2018.06.005.

Reviewer 3 Report

Abstract: Please check does. ‘1.96 g of chestnut tannins/kg of diet’ is not ‘1,96 g of chestnut tannins/kg of diet’.

Line 105. The clinical criterion of health or no toxicity signs in dairy cow need be described.

Statistical analyses: For the univariate data, the statistical evaluation by using only unpaired two-tailed Student’s t-test is inadequate: time and treatment, as well as the interaction therefrom must be considered first as fixed effects in an adequate model, before you test group differences within time points. The results presented in the present form (Figure 1 and 2) may substantially change, and thus their interpretation throughout the manuscript.

Results: Only analyze the changes of thyroid hormone and antioxidative variables in serum, which it not explains the antioxidant defence mechanisms interacting with thyroid status. So, correlation analysis between thyroid status and antioxidative capacity should be performed using Pearson or Spearman coefficient.

Results: Transition dairy cows usually undergoes the oxidative stress after calving. However, the antioxidant status during prepartum period has been analyzed in this study, why not analyze the effect of chestnut tannin on antioxidative capacity in dairy cows during postpartum period?

Please check carefully the references, such as ‘Abdollahi, M.; Ranjbar, A.; Shadnia, S.; Nikfar, S.; Rezaie, A. Pesticides and Oxidative Stress: A Review RA’ (line 419), the journal name and publication date missed.

Author Response

Comments and Suggestions for Authors

Abstract: Please check does. ‘1.96 g of chestnut tannins/kg of diet’ is not ‘1,96 g of chestnut tannins/kg of diet’.

Answer:  We apologize for this typo error. We corrected it in the revised version of the paper (please see line).

Line 105. The clinical criterion of health or no toxicity signs in dairy cow need be described.

Answer: We are thankful for this suggestion. We provide the clinical criterion of health or no toxicity signs in accordance with the suggestion:

“ The cows were clinically examined by general condition and only healthy cows with-out history of metabolic disorders in previous lactation were chosen for the study. The animals exhibited no clinical health problems or signs of tannin toxicity during the close-up period. The clinical visit was carried out on the day before first blood sam-pling (25 days before expected parturition) and it included general condition and BCS. During the trial period feed intake and feces consistency were monitored each day due to possible antinutritional effects of tannins. The cows were clinically examined on each day of study, by general condition as well as by signs of hepatotoxicity (in-apetance, lethargy and brisket oedema) and nephrotoxicity (oliguria, perirenal edema, azotemia and proteinuria).”

 (please see lines 137-145 in the revised version of the manuscript).

Statistical analyses: For the univariate data, the statistical evaluation by using only unpaired two-tailed Student’s t-test is inadequate: time and treatment, as well as the interaction therefrom must be considered first as fixed effects in an adequate model, before you test group differences within time points. The results presented in the present form (Figure 1 and 2) may substantially change, and thus their interpretation throughout the manuscript.

Answer: We apologize if the statistical analysis was not the clearest explaned in the manuscript. We use Factorial ANOVA test and based on ANOVA results t-test as a post hoc test for comparison between the tested groups. Also, we added a comparison within groups at d 25 and d 5 before expected parturition, and for this we used a dependent t-test.  In accordance with the suggestion of the reviewer, we clarified statistical methods we used (please see lines 200-208 in the revised version of the manuscript). Moreover, in the revised versions of the Figure 1 and 2, P values were provided for all statistically significant changes (or tendency toward significance) both within groups (d 25 vs. d 5 before expected parturition) and  comparison between the groups (CNT vs. CON group).

Results: Only analyze the changes of thyroid hormone and antioxidative variables in serum, which it not explains the antioxidant defence mechanisms interacting with thyroid status. So, correlation analysis between thyroid status and antioxidative capacity should be performed using Pearson or Spearman coefficient.

Answer: The initial statistical analysis we obtained revealed no significant correlation among thyroid hormones and antioxidative defence parameters in cows supplemented with chestnut tannins or unsupplemented cows. Nevertheless the significant correlation between T3 and SOD as important parameter of antioxidative defence  is often reported in hypothyroidism due to important role of thyroid hormones in lipid metabolism and antioxidant function of PON 1 (Baskol et al., 2007; Al-Naimi et al., 2018). However, we can speculate that adequate thyroid status of the cows in both groups might be the possible reason for lack of correlation between T3 and SOD, as supported by the observation of Al-Naimi et al. (2018), who reported that the correlation between thyroid hormones and PON 1 activity decline after the treatment of hypothyroidism. Although the positive correlation provides no indication of cause and effect, we consider that increased serum SOD activity in CNT cows in comparison to CON cows may be due to reduction in lipid peroxidation rather than stimulation of the synthesis of the enzyme protein.

Baskol, G.; Atmaca, H.; Tanrıverdi, F.; Baskol, M.; Kocer, D.; Bayram, F. Oxidative Stress and Enzymatic Antioxidant Status in Patients with Hypothyroidism before and after Treatment. Exp. Clin. Endocrinol. Diabetes 2007, 115, 522–526, doi:10.1055/s-2007-981457.

Al-Naimi, M.; Hussien, N.; Rasheed, H.; Al-kuraishy, H.; Al-Gareeb, A. Levothyroxine Improves Paraoxonase (PON-1) Serum Levels in Patients with Primary Hypothyroidism: Case–Control Study. J. Adv. Pharm. Technol. Res. 2018, 9, 113, doi:10.4103/japtr.JAPTR_298_18.

Results: Transition dairy cows usually undergoes the oxidative stress after calving. However, the antioxidant status during prepartum period has been analyzed in this study, why not analyze the effect of chestnut tannin on antioxidative capacity in dairy cows during postpartum period?

Answer: We measured chestnut tannins effect prepartaly since we previously established a positive effect of chestnut tannins on the metabolic and antioxidant status of cows during the same period of testing. As the obtained metabolic effects are similar to those caused by the thyroid hormones, we try to examine whether these effects can be related to thyroid activity under the influence of tannins, bearing in mind that some plant-based preparations that contain tannins have shown a stimulating effect on the activity of the thyroid gland. We wanted to avoid the period of early lactation, which is known to have low deiodinase activity, so we would have to use higher doses of chestnut tannins, which are known to have an anti-nutritive effect when given in high doses. We also wanted to avoid interrelationships between thyroid and growth hormones during the peripartal period. Based on the fact that marked changes in the thyroid activity occur in the close up period and that this activity can be impaired in a specific conditions  (such as heat stress), we focused our research on this period.  The disccusion related to the influence of heat stress on thyroid activity existed in unrevised version of our paper. (please see lines 222-224 in the un revised version of the manuscript).

In addition to the mentioned above, we also added a sentence in the discussion section which clarified significance of our study in the prepartal period:

“We employed a dietary approach during the close-up period with the main reason to avoid higher doses of chestnut tannins which may have an anti-nutritive effect. Namely, low deiodinase activity during the early lactation period would require a higher doses of chestnut tannins”

(please see lines 375-379 in the revised version of the manuscript)

Please check carefully the references, such as ‘Abdollahi, M.; Ranjbar, A.; Shadnia, S.; Nikfar, S.; Rezaie, A. Pesticides and Oxidative Stress: A Review RA’ (line 419), the journal name and publication date missed.

Answer: We checked references and corrected in this place and throughout the text (please see lines 524-525 of the revised version of the manuscript).

Reviewer 4 Report

General Comments to the Authors.
The study aim to investigate the effects of supplementation of chestnut on
antioxidant capacity and tried to prove that thyroid status is the main contributor. Overall, the manuscript is well written, although the reviewer is concerning about the data quality (the measurements of biochemicals are not novel and some of them have been previous used by the authors) of this manuscript did not meet the requirement of this journal. The other major concern I currently have include:

 1) the use of statistics with issues including repeated measures analysis does not appear to have been included even though the nature of the data requires a repeated measures analysis. With no clear statistics, doubts are easily raised on interpretation of results;

2) I am ok with reuse of the animals and samples which have been published previously, but the design of the manuscript has to be different to the previous publication. At least not to measure the antioxidant capacity again which occupied half of the present manuscript.

Author Response

The study aim to investigate the effects of supplementation of chestnut on antioxidant capacity and tried to prove that thyroid status is the main contributor. Overall, the manuscript is well written, although the reviewer is concerning about the data quality (the measurements of biochemicals are not novel and some of them have been previous used by the authors) of this manuscript did not meet the requirement of this journal. The other major concern I currently have include:

Answer: It is possible that by referring to our previous studies we have confused the reviewers. The presented study is completely new and none of the data and results were used from our previous studies. In our previous studies, we were focused on examining the influence of chestnut tanins on the parameters of antioxidant defense and metabolic parameters in cows in the puerperal period. Bearing in mind the importance of the regulation of thyroid hormones in this period as well as the possible connection between the mechanisms of antioxidant defence and thyroid activity, we hypothesized that these mechanisms may be affected by chestnut tannin supplementation.

According to the suggestion of the rewiewer, our hypothesis is clearly defined in the revised version of the paper (please see lines 115-117 of the revised version of the paper), but the introduction has also been rearranged in such a way as to more clearly indicate the objective of our current research (please see lines 46-94 of the revised version of the paper). Also some parts that referred to our previous studies were removed from the revised text in order to avoid confusing the reader of our manuscript.

(lines  34, 94-96 from non revised version of the manuscript are excluded).

 1) the use of statistics with issues including repeated measures analysis does not appear to have been included even though the nature of the data requires a repeated measures analysis. With no clear statistics, doubts are easily raised on interpretation of results;

Answer: We apologize if the statistical analysis was not the clearest explaned in the manuscript. We use Factorial ANOVA test and based on ANOVA results t-test as a post hoc test for comparison between the tested groups. Also, we added a comparison within groups at d 25 and d 5 before expected parturition, and for this we used a dependent t-test.  In accordance with the suggestion of the reviewer, we clarified statistical methods we used (please see lines 200-208 in the revised version of the manuscript). Moreover, in the revised versions of the Figure 1 and 2, P values were provided for all statistically significant changes (or tendency toward significance) both within groups (d 25 vs. d 5 before expected parturition) and  comparison between the groups (CNT vs. CON group).

2) I am ok with reuse of the animals and samples which have been published previously, but the design of the manuscript has to be different to the previous publication. At least not to measure the antioxidant capacity again which occupied half of the present manuscript.

Answer: As stated in one of the previous comments the presented study is completely new and none of the data and results were used from our previous studies. Starting from our previous research, we hypothesized that the underlying mechanisms of antioxidant defence that are altered in tannin supplementation are linked to thyroid activity, which has not been proven so far in studies on cattle. In order to prove this, a new study was designed, but due to the connection between antioxidant defense and thyroid activity, we had to repeat part of the analyzes related to the parameters of antioxidant defense.  However, taking into account the reviewers' suggestion, the introduction was changed in such a way as to focus on the influence of tannin supplementation on thyroid activity (please see lines 73-102 in the revised version of the manuscript). Also, the discussion has been significantly changed in its initial part and the part related to the mechanisms of antioxidant protection has been excluded (please see lines 241-316 in the revised version of the manuscript). In this way, we focused on the primary goal of our study and avoided repetition in the introduction and discussion.

Reviewer 5 Report

Overall, this manuscript contributes original metabolic research. It is quite well-written, clear, relevant for the ruminant metabolism and presented in a well-structured manner. The aim of the study was clear. Although data in relation to thyroid function and antioxidant systems in the cow model are limited, the authors did a good job of the connection to previous/relevant studies and potential explanations in the discussion.

However, specific comments are minor and related to the content in Materials and Methods, Discussion & Conclusion, which could be used for fine tuning.

1. Line 109: The authors designate the -5d as day 5 prior to expected parturition. How many (or percentage) dairy cows truly calved or did not calve as expected in the present study? Could this issue affect the metabolites and thyroid hormone levels, oxidative stress indicators or antioxidant activities? I would add these to the results and/or discussion.

2. The authors mention the strengths of this study; however, limitations are missing. I would add this to the discussion.

3. Line 231-232: According to the statement “The results of decreased AST and LDH activities in CNT cows further support strong antioxidant properties of chestnut tannins.”, please clearly explain how AST and LDH activities found in this study support the antioxidant properties of chestnut tannins. Please clearly explain.    

4. Lines 323-325: In conclusion, according to the statement “Further research into the implications of such a dietary approach to reproductive efficiency will be needed since the carryover effects can persist into early lactation.”, what does “…..the carryover effects can persist into early lactation.” mean? Please clearly state.   

5. For the implications of a dietary approach for further research, it would be more beneficial to also investigate at least in postpartal cows, where thyroid hormones often alter and oxidative stress is more pronounced.

Author Response

Overall, this manuscript contributes original metabolic research. It is quite well-written, clear, relevant for the ruminant metabolism and presented in a well-structured manner. The aim of the study was clear. Although data in relation to thyroid function and antioxidant systems in the cow model are limited, the authors did a good job of the connection to previous/relevant studies and potential explanations in the discussion.

However, specific comments are minor and related to the content in Materials and Methods, Discussion & Conclusion, which could be used for fine tuning.

  1. Line 109: The authors designate the -5d as day 5 prior to expected parturition. How many (or percentage) dairy cows truly calved or did not calve as expected in the present study? Could this issue affect the metabolites and thyroid hormone levels, oxidative stress indicators or antioxidant activities? I would add these to the results and/or discussion.

Answer:  We are thankful for this suggestion  and we also think this is an important issue.  In the non-revised version of our manuscript, we did not explain this in detail, but in fact, after the initial introduction to the experiment, which was expressed 25 days before the expected calving date and shown through standard deviations (25±2 d), blood was drawn every day before calving, and analyzes were made from the samples on  day 5 before calving. We chose this approach precisely for the reason that  the parameters can vary significantly and change from on daily basis before calving.  In order to meet this reviewers' suggestion we we have modified the material and methods section related to this question and clarified the methodology (please see lines 149-151 in the revised version of the manuscript).

  1. The authors mention the strengths of this study; however, limitations are missing. I would add this to the discussion.

Answer:   In order to meet the reviewer's suggestion, we added discussion that highlights limitation of our study. Namely, we employed a dietary approach during the close-up dry period with the main reason to avoid the interrelationships between thyroid and growth hormones during the peripartal period. Reactive oxygen species (ROS) are also known that may inhibit thyroid hormone production (Kochman et al., 2021). The half-life of ROS is very short and therefore we analyzed antioxidative enzymes and a product of lipid peroxidation instead. In addition, hepatic deiodinase activity is an important control point for regulating the thyroid status in various physiological and pathological conditions (Meyerholz et al., 2016). Thus, it would be interesting to examine the effect of chestnut tannins on hepatic deiodinase gene expression to further dissect the cause of improved thyroid status in prepartum cows. (please see lines 369-375 in the revised version of the manuscript).

Kochman, J.; Jakubczyk, K.; Bargiel, P.; Janda-Milczarek, K. The Influence of Oxidative Stress on Thyroid Diseases. Antioxidants 2021, 10, 1442, doi:10.3390/antiox10091442.

Meyerholz, MM.; Mense, K.; Linden, M.; Raliou, M.; Sandra, O.; Schuberth, H-J.; et al. Peripheral thyroid hormone levels and hepatic thyroid hormone deiodinase gene expression in dairy heifers on the day of ovulation and during the early peri-implantation period. Acta Vet Scand. 2015; 58, 52, doi.org/10.1186/s13028-016-0231-6.

  1. Line 231-232: According to the statement “The results of decreased AST and LDH activities in CNT cows further support strong antioxidant properties of chestnut tannins.”, please clearly explain how AST and LDH activities found in this study support the antioxidant properties of chestnut tannins. Please clearly explain.

Answer: In order to meet the reviewer's suggestion, we rearrange discussion section and place this sentence later in the text where the connection of reactive oxygen species, lipid peroxidation and cell damage is explained. We have further explained the sentence so that is now stated:

“Since the increased extracellular activities of AST and LDH could be the caused of disrupted cell membrane integrity, which occurs during lipid peroxidation, under the influence of free radicals lower  AST and LDH activities in CNT cows in our research further support strong antioxidant properties of chestnut tannins [42].”

(please see lines 352-356 in the revised version of the manuscript)

  1. Lines 323-325: In conclusion, according to the statement “Further research into the implications of such a dietary approach to reproductive efficiency will be needed since the carryover effects can persist into early lactation.”, what does “…..the carryover effects can persist into early lactation.” mean? Please clearly state.

Answer: We measured chestnut tannins effect prepartaly since we previously established a positive effect of chestnut tannins on the metabolic and antioxidant status of cows during the same period of testing. As the obtained metabolic effects are similar to those caused by the thyroid hormones, we try to examine whether these effects can be related to thyroid activity under the influence of tannins, bearing in mind that some plant-based preparations that contain tannins have shown a stimulating effect on the activity of the thyroid gland. We wanted to avoid the period of early lactation, which is known to have low deiodinase activity, so we would have to use higher doses of chestnut tannins, which are known to have an anti-nutritive effect when given in high doses. We also wanted to avoid interrelationships between thyroid and growth hormones during the peripartal period. Based on the fact that marked changes in the thyroid activity occur in the close up period and that this activity can be impaired in a specific conditions  (such as heat stress), we focused our research on this period.  The disccusion related to the influence of heat stress on thyroid activity existed in unrevised version of our paper (please see lines 222-224 in the non revised version of the manuscript). In the desire to meet the reviewer's suggestion we redefine the conclusion so it clearly indicates the period of tannin supplementation in our research so it now states:

“The use of chestnut tannins in close-up dry cows has the potential to improve thyroid function near parturition. Therefore, dietary supplementation with chestnut tannins is relevant not only for improving the antioxidant status but also for the achievement of proper thyroid function in dairy cows. Further research into the implications of such a dietary approach in formulating rations in close-up period  will be needed, especially in specific conditions of altered thyroid function (such as heat stress).”

  1. For the implications of a dietary approach for further research, it would be more beneficial to also investigate at least in postpartal cows, where thyroid hormones often alter and oxidative stress is more pronounced.

Answer: Please see the answer above.  In addition to the mentioned above, we also added a sentence in the discussion section which clarified significance of our study in the prepartal period:

“We employed a dietary approach during the close-up period with the main reason to avoid higher doses of chestnut tannins which may have an anti-nutritive effect. Namely, low deiodinase activity during the early lactation period would require a higher doses of chestnut tannins”

(please see lines 375-379 in the revised version of the manuscript).

Round 2

Reviewer 3 Report

The paper should be published. 

Reviewer 4 Report

The reviewer thanks the authors for their contribution to revise the manuscript and their responses to my comments.

However, the main issues I pointed out have not been improved properly.

For example, you have a time course evaluation for these data. Your observations are not independent from each other in a time-series and a repeat measures analysis with an appropriate covariance structure should be used. I believe you must run your data again with a model that includes the fixed effect of day with an appropriate covariance structure in a repeated measures model.

For the reuse of data for publication, the author explained that they want to study the possible mechanism, mainly focusing on thyroid activity, for how does chestnut tannin affect the antioxidant defence. The data that was generated from the present manuscript (the results of metabolites related to funciton of thyroid) was not good enough for a paper to be considered to publish in Metabolites.